# Geographic weighted regression analysis of hot spots of modern contraceptive utilization and its associated factors in Ethiopia

Yazachew Moges Chekol[1]*, Setotaw Begashaw Jemberie[2], Bazezew Takel Goshe[3], Getayeneh Antehunegn Tesema[3], Zemenu Tadesse Tessema[3], Lewi Goytom Gebrehewet[3]

1 Department of Health Information Technician, Mizan Aman College of Health Science, Mizan Aman, Ethiopia, 2 Central Gondar Zone Health Department, Amhara Regional State Health Bureau, Bahir Dar, Ethiopia, 3 Department of Epidemiology and Biostatistics, Institute of Public Health, College of Medicine and Health Sciences, University of Gondar, Gondar, Ethiopia

* yazachewmoges@gmail.com

**Data Availability Statement:** All relevant data are within the manuscript and its Supporting Information files.

## Abstract

### Background

Utilization of modern contraceptives is a common healthcare challenge in Ethiopia. Prevalence of modern contraception utilization is varying across different regions. Therefore, this study aimed to investigate Geographic weighted regression analysis of hotspots of modern contraceptive utilization and its associated factors in Ethiopia, using Ethiopian Demographic and Health Survey 2016 data.

### Methods

Based on the 2016 Ethiopian Demographic Health Survey data, a total weighted sample of 8,673 women was included in this study. For the Geographic Weighted Regression analysis, Arc-GIS version 10.7 and SaTScan version 9.6, statistical software was used. Spatial regression was done to identify factors associated with the hotspots of modern contraceptive utilization and model comparison was carried out using adjusted $R^2$ and AICc. Variables with a p-value < 0.25 in the bi-variable analysis were considered for the multivariable analysis. Multilevel robust Poisson regression analysis was fitted for associated factors since the prevalence of modern contraceptive was >10%. In the multilevel robust Poisson regression analysis, the adjusted prevalence ratio with the 95% confidence interval was reported to declare the statistical significance and strength of association.

### Result

The prevalence of modern contraceptive utilization in Ethiopia was 37.25% (95% CI: 36.23%, 38.27%). Most of the hotspot areas were located in Oromia and Amhara regions, followed by the SNNPR region and Addis Ababa City administration. Single Women, poor Women, and more fertility preference were significant predictors of hotspots areas of modern contraceptive utilization. In the multivariable multilevel robust Poisson regression analysis, Women aged 25–34 years (APR = 0.88, 95% CI: 0.79, 0.98), 35–49 years (APR = 0.71, 95% CI: 0.61, 0.83), married marital status (APR = 2.59, 95% CI: 2.18, 3.08), Others

**Funding:** The author(s) received no specific funding for this work.

**Competing interests:** The authors have declared that no competing interests exist.

**Abbreviations:** AGYW, Adolescent Girls and Young Women; AICc, Corrected Akaike Information Criteria; CPR, Contraceptive Prevalence Rate; EDHS, Ethiopian Demographic Health Survey; GWR, Geographic Weighted Regression; HDI, Human Development Index; MCPs, Modern Contraceptives; TCPS, Traditional Contraceptives.

religions (APR = 0.76, 95% CI: 0.65, 0.89), number of children 1–4 (APR = 1.18, 95% CI: 1.02, 1.37), no more fertility preference (APR = 1.21, 95% CI: 1.11, 1.32), Afar, Somali, Harari, and Dire Dawa: (APR = 0.42, 95% CI: 0.27, 0.67), (APR = 0.06, 95% CI: 0.03, 0.12), (APR = 0.78, 95% CI: 0.62, 0.98), and (APR = 0.75, 95% CI: 0.58, 0.98), respectively. Amhara region (APR = 1.34, 95% CI: 1.13, 1.57), rural residence (APR = 0.80, 95% CI: 0.67, 0.95) High community wealth index (APR = 0.78, 95% CI: 0.67, 0.91) were significantly associated with modern contraceptive utilization.

## Conclusion and recommendation

There were significant spatial variations of factors affecting modern contraceptive use across regions in Ethiopia. Therefore, public health interventions targeting areas with low modern contraceptive utilization will help to increase modern contraception use considering significant factors at individual and community levels. The detailed map of modern contraceptive use cold spots among reproductive age group and its predictors could assist program planners and decision-makers to design targeted public health interventions. Government of Ethiopia must develop more geographic targeted strategies for improving socioeconomic status of women and availability & accessibility of health facilities in rural areas of the countries.

## Background

Generally contraceptive methods are broadly classified as modern contraceptives (CPs) and traditional contraceptive (TC) methods. Modern contraceptive have been identified as an effective method for fertility reduction, and are thus being widely promoted to slow rapid population growth, particularly in developing countries [1, 2]. Promoting access to modern contraceptives among women of reproductive age has also proven to be an effective public health intervention to improve maternal and child health outcomes [3, 4]. Worldwide, modern contraceptives are important in fertility contro l [5]. In low income countries, utilizing modern contraceptives have a clear effect on the health of women, children and families. For instance, contraceptives are estimated to prevent 2.7million infant deaths and the loss of 60million healthy lives a year worldwide [6]. Promoting contraceptives in nations with high birth rates prevents 32% of all maternal fatalities and roughly 10% of infant deaths. Modern contraceptives also make a huge contribution to the achievement of universal primary schooling, female empowerment, and in reducing poverty and hunger [7]. Family planning is also important in preventing unintended pregnancies and unsafe abortions [8, 9].

Despite its importance, access to and utilization of modern contraceptives vary worldwide. Women in developed countries have better access to and use of contraceptives compared with women in developing countries [8]. In a study from 2010 to 2014, it was reported that the global burden of unintended pregnancies was 44%; the rate of unintended pregnancies is substantially higher in developing countries compared with developed regions [10]. There is high unmet need for modern contraception in low income countries and this may contribute to higher rates of unintended pregnancies. For instance, in sub-Saharan Africa, the prevalence of contraceptive use among women of reproductive age is only 17% [11].

Similarly, the utilization of modern contraceptives is a common healthcare challenge in Ethiopia [12]. Prevalence of modern contraception use is varied across different regions. For instance, the Somali region accounts for the lowest rate of modern contraceptive use (1.4%), compared with Addis Ababa (50.1%) [12].

Whether they are married or not, the use of modern contraceptive among adolescent girls and young women (AGYW) had very low compared to other age groups in the developing world. This means, the contraceptive need of the AGYW deserve further international interventions [13]. Regardless of the introduction of modern contraception method over the last some decades, the level of utilization was also not adequate among all reproductive age women. In Ethiopia only 36% of married women in reproductive age (15–49 years) used modern contraceptives [12]. By giving attention for this, the advantage of the modern contraceptive use among reproductive age is very vital for designing interventions, plans, and policies to address premature age pregnancies and other related issues. It is also useful to reduce unsafe abortions, maternal death, and sexually transmitted infections (STI). Low use of modern contraceptives among reproductive age group is the result of several contributing factors [14, 15].

In Ethiopia, Modern contraceptives are provided without charge in each government health facility to encourage utilization. Determinant factors associated with modern contraception method utilization are; number of living children, women's current age, age at first birth, education, marital status, terminated pregnancy, religious affiliation, media exposure about family planning, wealth index, working status, fertility preference, and distance to health facility, are the main determinant factors for utilization of modern contraceptives [16, 17]. Community level factors are a variable that are shared by the community which affects either positively or negatively as a community rather than individual level and major reason to consider multilevel analysis of this study. Community-level factors were found to be associated with contraceptive utilization are; region, community level wealth index, place of residence and community level media exposure were community-level determinants in Nigeria and Mozambique [18, 19].

The previous study was conducted onlyconsidering married women as a study population and without considering of geographical variationofvariablesacross Ethiopian regions.But there is evidence that indicates variation of variables influence among regions in health service utilization and unmarried women aresexually active almost the same as married women [20]. Therefore, this study aimed to investigate spatial regression analysis of modern contraceptives use among women of reproductive age in Ethiopia, irrespective of their marital status and identify the potential factors associated with the use of modern contraceptives and considering geographic variation of variables. As a result this study result will facilitate evidence based decision making by complementinglimitations of the previous studies.

The findings of this study will be useful for health planners, policymakers, and non-governmental partners who are working to improve the health and well-being of women of reproductive age in Ethiopia. Besides, mapping hotspot areas of modern contraceptive use, it will provide a deeper understanding of the impacts of already implemented interventions in each region of the country. Furthermore, it will assist in designing programs and strategies to increase coverage, quality, and equity of women's reproductive health at the country level.

## Methods and materials

### Study design, data source and period

A population based cross sectional study was conducted from January 18, 2016, to June 27, 2016 using EDHS 2016 data set. The survey data was accessed from the measure Demographic and Health Survey (**https://dhsprogram.com/**).

### Study area

The EDHS 2016 is fourth nationwide survey conducted in Ethiopia. Ethiopia is the second populous country in Africa next to Nigeria with a population of more than one hundred million. Administratively, Ethiopia is divided into nine geographical regions (Tigray, Afar,

Amhara, Oromia, Somali, Benishangul-Gumuz, SNNPR, Gambella and Harari) and two administrative cities, Addis Ababa and Dire Dawa. Ethiopia shares the boundaries in the North with Eritrea, in the South with Kenya and Somalia, in the West with South Sudan and North Sudan, in the East with Djibouti and Somalia. The modern contraceptive methods are freely available without any fee for reproductive-age women, which are provided in all public health facilities in Ethiopia.

## Source and study population

**Source population.** The source population was all Women in the reproductive age (15–49 years) in Ethiopia.

**Study population.** The study population was all Women in the reproductive age (15–49 years) in Ethiopia from January 18, 2016, to June 27, 2016.

## Inclusion and exclusion criteria

**Inclusion criteria.** All Women in the reproductive age group were included in the study.

**Exclusion criteria.** Women never had sex, pregnant women and Enumeration Areas (EA) with zero longitude and latitude were excluded.

## Sampling procedures

In this study, interviews were completed for 15,683 women. A total of 8,673 eligible women were included after the necessary exclusion criteria were carried out (Fig 1). A Woman was asked whether they had obtained any modern contraceptive method in the 5 years preceding the survey. Weighted number was used to restore the representative of sample data.

## Study variables

**Outcome variable.** Women received any modern contraceptive method (yes/no)

**Independent variables.** Individual level factors included were age, religion, marital status, working status of women, education status, wealth index, number of living children, terminated pregnancy, age at first birth, media exposure, fertility preference, and distance to health facility.

Community level factors include community wealth index, residence (urban and rural), region, community media exposure.

The shape files of Ethiopia regional, zonal, district level shape files were obtained from the Central Statistical Agency of Ethiopia.

## Operational definitions

**Modern contraception method:** A women/men was considered to be using modern contraception if she/he used any of the following modern contraceptive methods: female/male sterilization, contraceptive pills, IUD, injectable, implants (norplants), diaphragm, lactational amenorrhea method(LAM), standard days method(SDM), emergency contraception and male/female condom [21].

**Community wealth index:** This was generated by aggregating household wealth index at cluster/EA level. It was categorized as high community poverty when the proportion of women whose wealth index below the national median value (.3333333) and low community poverty when the proportion of women whose wealth index above the median value (.3333333) was lower than the median values as this variable was not normally distributed.

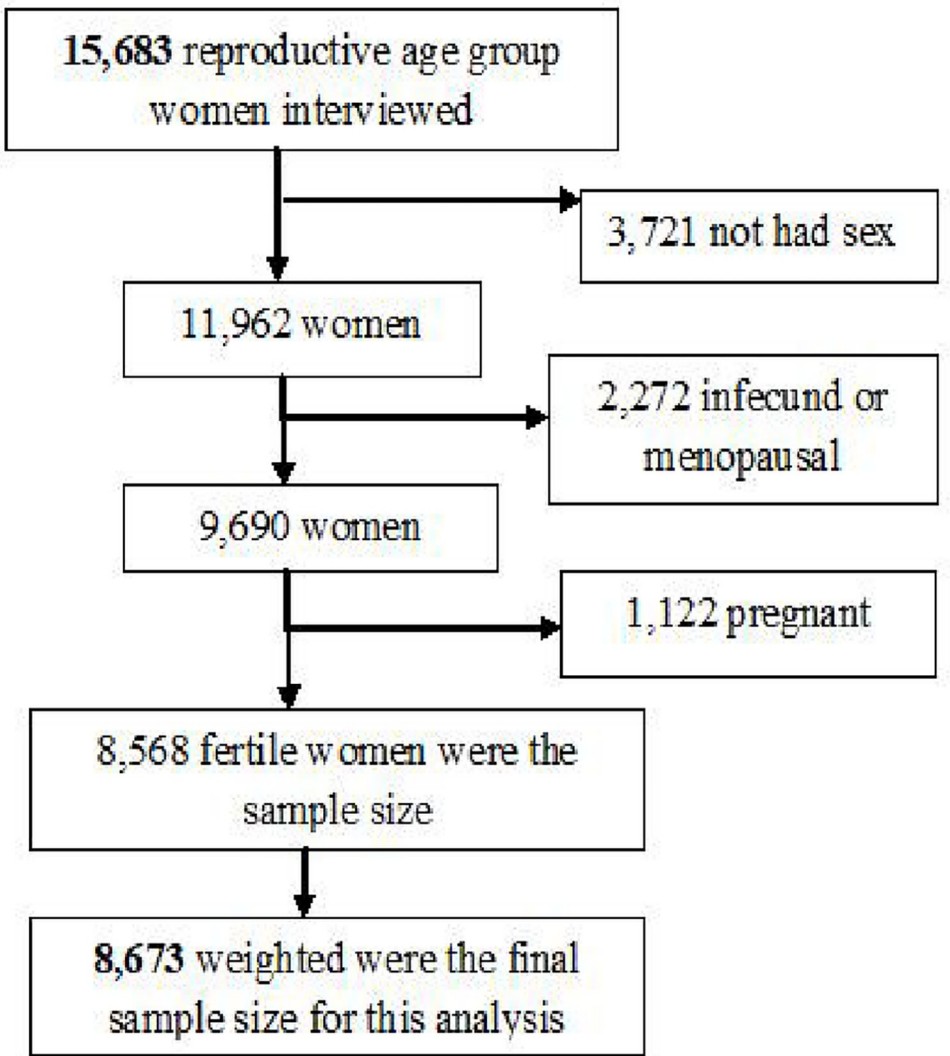

**Fig 1. Schematic illustration of women included in the study.**

**Media exposure**: is defined as household who had listening to radio or watching television or reading newspaper/magazine at least once a week were considered as exposed to media [12].

**Community media exposure**: if proportion of women who had listing to radio, watching television, and reading newspaper/magazine was below the median were considered as "low media exposure" (.1666667) and above the median were considered as "high media exposure" (.1666667).

**Hot spot:** Areas with high modern contraception utilization

**Cold spot:** Areas with low contraception utilization

## Data management and analysis

Data extraction, coding, and analysis were done using STATA version 16, Arc-GIS version 10.7 and SaTScan version 9.6 statistical software's. Socio-demographic characteristics of the study participants were calculated in frequency and percentage. To re-establish the

representativeness of the data weighted data was used for analysis. To consider the hierarchical nature of EDHS data multilevel Poisson regression analyses were employed and clustering effect was assessed using Intra-class Correlation Coefficient (ICC),clustering effect was considered because of ICC >10%. In this case, the prevalence ratio is the best measures of association to minimize over estimation of association between outcome and independent variables. Variables with a p-value of < 0.25 at the bi-variable multilevel Poisson regression analysis were included into the final model of multivariable regression analysis model, in which prevalence ratio with 95% confidence intervals were estimated to identify independent variables of modern contraceptive use. To declare statistical significance, p-values less than 0.05 were used.

## Spatial analysis

The spatial autocorrelation (Global Moran's I) statistics was used to evaluate whether the modern contraception method utilization distribution is random or not at the national level. Moran's, I value close to −1 indicates that modern contraception method utilization is dispersed, whereas Moran's I close to +1 indicates modern contraception method utilization is clustered and if Moran's I close to 0 revealed that modern contraception method utilization is randomly distributed. A statistically significant Moran's I ($p < 0.05$) value showed that modern contraception method utilization is non-random [22, 23].

Hot spot analysis was computed to measure how spatial autocorrelation varies over the study location by calculating Getis-ordGi* statistic for each area. To determine whether there was substantial clustering, the Z-score and p-values were calculated. Statistical values with high Getis-ordGi* indicate "hotspot" whereas low Getis-ordGi* means a "cold spot". Empirical Bayesian kriging interpolation was used to predict modern contraceptive use in un-sampled areas.Spatial scan statistical analysis was used to perform cluster analysis to detect the more likely clusters by computing the relative risk (RR) and testing the statistical significance [24].

## Spatial regression

Ordinary Least Squares (OLS) regression; the spatial regression modeling was performed to identify predictors of the spatial heterogeneity of modern contraceptive method utilization. OLS is a global statistical model for testing and explaining the relationship between the outcome and explanatory variables [25]. Also used as a diagnostic tool for selecting the appropriate predictors for the Geographic Weighted Regression (GWR) model [26]. Different assumptions werecheckedbefore jumping to GWR analysis like non-stationarity, residual spatial autocorrelation, model bias and multicollinearity was assessed using Koenker test, Moran's I, Jarque-Bera statistics, and VIF respectively.

GWR is a local spatial statistical technique that assumes the non-stationarity in relationships between the outcome and predictors across EAs [26]. The GWR analysis was employed with evidenceofKoenker test statistically significant. In the GWR analysis, the coefficients of the predictors take different values across the study area. Mapping the GWR coefficients associated with the predictors and provides insight for targeted interventions. Best fitted model for the data was identified using corrected lowest AIC and higher adjusted $R^2$.

## Result

A total weighted sample of 8,673 reproductive age Women included. The mean age of the study participants was 29.54 (SD of ±7.73) years. More than half (44.13%) of the women were 25–34 age group. The vast majority (85.7%) of the women were married. More than one thirds (37.48%) of the respondent were in the Oromia region and about7043 (81.21%) were from the rural areas. Regarding maternal education, the majority (58.51%) of the respondents didn't

attain formal education. Nearly half (44.25%) of the respondents were Orthodox religion followers. Majority of the respondents (41.63%) were rich and more than half of the respondent (58.24%) were had more fertility preference (Tables 1 and 2).

## Prevalence of modern contraceptive utilization

The prevalence of modern contraceptive utilization in Ethiopia was 37.25% (95% CI: 36.23%, 38.27%). The utilization of modern contraceptives varied across different regions and city administrations; the highest modern contraceptive was reported in Oromia region (37.48%) followed by Amhara (25.6%) and SNNPR (18.57%) regions. The lowest prevalence of modern contraceptive utilization was observed in Harari (0.23%), Gambela (0.32%), and Afar (0.89%) regions.

## Spatial distribution of modern contraceptive utilization

A total of 623 clusters were considered for the spatial analysis of modern contraceptive utilization, Points on the map represent the clusters and their corresponding proportion of modern contraceptive utilization. The red color indicates areas with low proportion of modern contraceptive utilization whereas the blue color represents areas with high proportion of modern contraceptive utilization. The highest prevalence of modern contraceptive utilization among reproductive age women was observed in Oromia, Amhara, Addis Ababa, and SNNPR (Fig 2).

## Spatial autocorrelation of modern contraceptive utilization

The spatial distribution of modern contraceptive use among reproductive age Women in Ethiopia was non-random (Global Moran's I = 0.32; Z-score = 19.78; p-value 0.0000) (Fig 3).

## Hotspot analysis of modern contraceptive utilization

Most of the hot spot areas (blue color), those with high modern contraceptive prevalence rates, were located in Oromia and Amhara region, followed by the SNNPR region and Addis Ababa city administration. On the other hand, the majority of the cold spot areas (red color), those with low modern contraceptive prevalence rates, were located in the Gambela, Somali, and Afar regions followed by Tigray. This clustering was supported by the Getis-ordGi* statistic when conducting the spatial analysis (Fig 4).

## Spatial interpolation of modern contraceptive utilization

Using the Empirical Bayesian kriging interpolation, the green ramp color on the map indicates the predicted highest modern contraceptive utilization rates in Amhara, Tigray, Afar, Oromia, Northern Somali and Northern BeneshanguGumuz regions. However, the dim red color indicates low modern contraceptive utilized areas predicted in North and Southern Somali, Addis Ababa, Southern BeneshangulGumuz, Northern Afar and south and Northern Oromia (Fig 5).

## Spatial scan statistical analysis

The spatial scan statistics identified a total of 211 high, medium and low performing spatial clusters of modern contraceptive utilization. Of these, 180 clusters were most likely primary clusters (high performing clusters) accounting for 46.7%, 15 were secondary clusters (medium performing clusters) accounting for 64.9% and 16 were third clusters (low performing clusters) accounting for 64.3%. The green colors (rings) indicate the most statistically significant spatial window which contains primary clusters located in the Amhara, Addis Ababa, and Benshagul-Gumuz regions. This was centered at 10.575333 N, 37.480816 E with 293.34 km radius, with a

**Table 1. Socio-demographic characteristics of women, EDHS 2016.**

| Individual level characteristics of respondents in Ethiopia, 2016 (n = 8,673) | | |
| --- | --- | --- |
| **Variables** | **Weighted frequency (n = 8673)** | **percentage (%)** |
| Age of respondents (years) | | |
| 15–24 | 2,266 | 26.12 |
| 25–34 | 3,827 | 44.13 |
| 35–49 | 2,580 | 29.75 |
| Educational status | | |
| No | 5,075 | 58.51 |
| Primary | 2,520 | 29.06 |
| Secondary and above | 1,078 | 12.43 |
| Marital status | | |
| Single | 1,240 | 14.3 |
| Married | 7,433 | 85.7 |
| Religion | | |
| Orthodox | 3838 | 44.25 |
| Protestant | 64 | 0.74 |
| Muslim | 1791 | 20.65 |
| Other* | 2980 | 34.36 |
| Age at first birth (n = 7,684) | | |
| < = 19 rears | 4929 | 64.15 |
| 20–24 years | 2184 | 28.42 |
| > = 25 years | 571 | 7.43 |
| Number of living children | | |
| 0 | 1,048 | 12.09 |
| 1_ 4 | 4,984 | 57.47 |
| > = 5 | 2,640 | 30.44 |
| Fertility preference | | |
| want another | 5051 | 58.24 |
| undecided | 493 | 5.68 |
| want no more | 3129 | 36.08 |
| Currently working status | | |
| no | 5,808 | 66.97 |
| yes | 2,864 | 33.03 |
| Wealth index combined | | |
| poor | 3,322 | 38.31 |
| middle | 1,740 | 20.06 |
| rich | 3,610 | 41.63 |
| Terminated pregnancy | | |
| no | 7841 | 90.41 |
| yes | 832 | 9.59 |
| Distance to health facility | | |
| big problem | 4,636 | 53.45 |
| not a big problem | 4,037 | 46.55 |
| Media exposure | | |
| low media exposure | 6,614 | 76.27 |
| high media exposure | 2,058 | 23.73 |

*Other: Catholic and traditional

**Table 2. Community level characteristics, EDHS 2016.**

| Community level characteristics of respondents in Ethiopia, 2016(n = 8,673) | | |
|---|---|---|
| **Variables** | **Weighted frequency (n = 8673)** | **percentage (%)** |
| Region | | |
| Tigray | 653 | 7.52 |
| Afar | 77 | 0.89 |
| Amhara | 2220 | 25.6 |
| Oromia | 3251 | 37.48 |
| Somali | 233 | 2.69 |
| Benishangul | 91 | 1.05 |
| SNNPR | 1610 | 18.57 |
| Gambela | 28 | 0.32 |
| Harari | 20 | 0.23 |
| Addis Ababa | 442 | 5.1 |
| Dire Dawa | 47 | 0.55 |
| Residence | | |
| urban | 1,629 | 18.79 |
| rural | 7,043 | 81.21 |
| Community wealth index | | |
| Low | 4,185 | 48.26 |
| high | 4,488 | 51.74 |
| Community media exposure | | |
| low media exposure | 4,938 | 56.94 |
| high media exposure | 3,735 | 43.06 |

relative risk (RR) of 1.53 and Log-likelihood ratio (LLR) of 114.68, at p-value <0.01. The secondary clusters (yellow rings) located in the Western part of Oromia. The third one is located in the Southwestern part of SNNPR. The second clusters spatial window was centered at 6.721839 N, 38.294189 E with 54.02 km radius, with a relative risk (RR) of 1.80 and Log-Likelihood ratio (LLR) of 59.91, at p-value <0.01. The third clusters (red rings) spatial window was centered at 7.645646 N, 35.353485 E with 66.80 km radius, with a relative risk (RR) of 1.74 and Log-Likelihood ratio (LLR) of 21.08, at p-value <0.01 (Fig 6)

## Ordinary least square (OLS) regression analysis

The OLS model was computed to diagnose multicollinearity between the independent variables and all variables were less than 7.5. In the OLS analysis, the model explained about 38% (adjusted $R^2$ = 0.38) of the variation in modern contraceptive utilization among women with AICc = -307.87. Koenker test is used to check the relationships in our model non-stationary or not, the Koenker statistics were statistically significant in our model, indicates that the relationship between the explanatory variables and the outcome variable was non-stationary or heterogeneous across the study areas. Since, Koenker statistics were significant robust probabilities was used to screen out significant predictors; proportion of single women, proportion of poor women, and proportion of women more fertility preference were significantly associated with the prevalence of modern contraceptive utilization among women in the OLS model (Table 3).

The Joint F-statistics and Wald statistics were highly significant (p<0.01), which proves that the model was statistically significant. The spatial distribution of residuals was not normally distributed as the JarqueBera statistics were statistically significant (0.016310*). Spatial

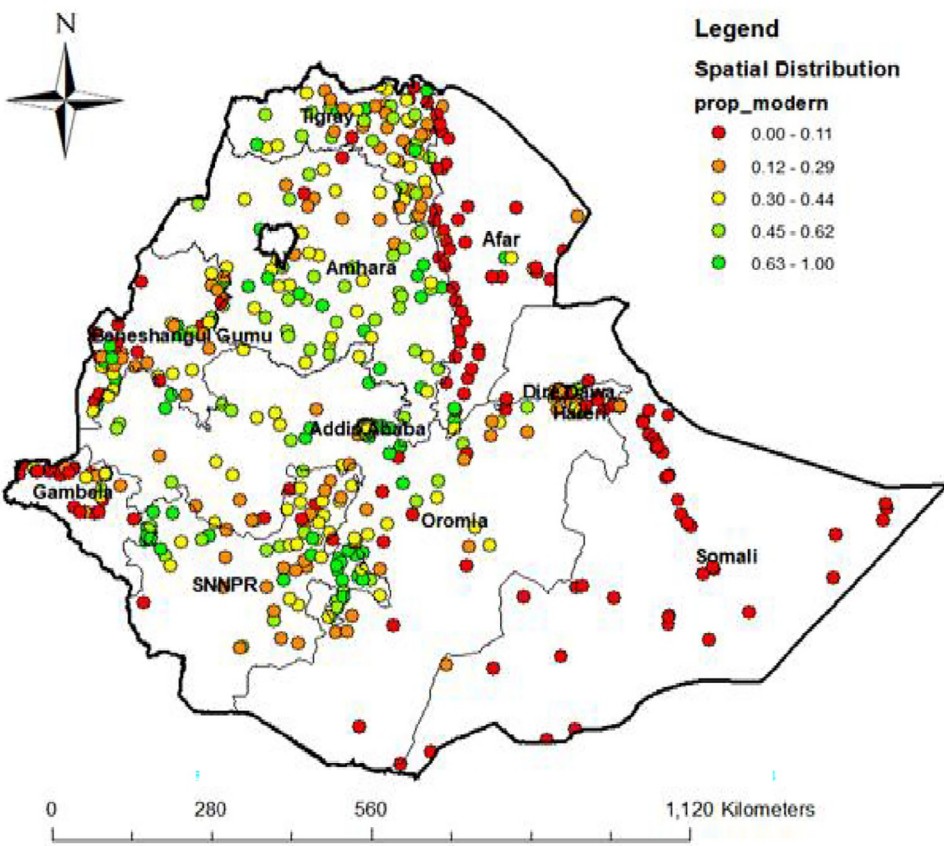

**Fig 2. Spatial distribution of modern contraceptive utilization in Ethiopia, 2016.** Source: Shape file from Ethiopia Central Statistical Agency (CSA).

autocorrelation was done and the residuals were not normally distributed (Moran's Index = 0.18) (p = 0.0000). This indicates that GWR should be applied. Since, the Koenker statistics showed the non-stationarity in the relationship as it assumes the spatial heterogeneity of the relationship between independent and dependent variables across space. The same numbers of independent variables were used for GWR analysis.

## Geographically weighted regression (GWR) analysis

The result of GWR analysis showed that there was a significant improvement over the global model (OLS). The AICc value decreased from -307.87 to -411.61. This implies that GWR best explains the spatial heterogeneity of modern contraceptive utilization among women, the difference was 103.74. In addition, the model's ability to explain modern contraceptive utilization has been improved 13% by using GWR analysis. Since, the adjusted $R^2$ was 0.51, (Table 4). In the geographically weighted regression analysis, the proportion of single women, the proportion of poor women, and the proportion of more fertility preference were significant predictors of hotspots areas of modern contraceptive utilization among women. The above three factors were considered as independent variables in the GWR analysis. Since, it was significant in the OLS analysis.

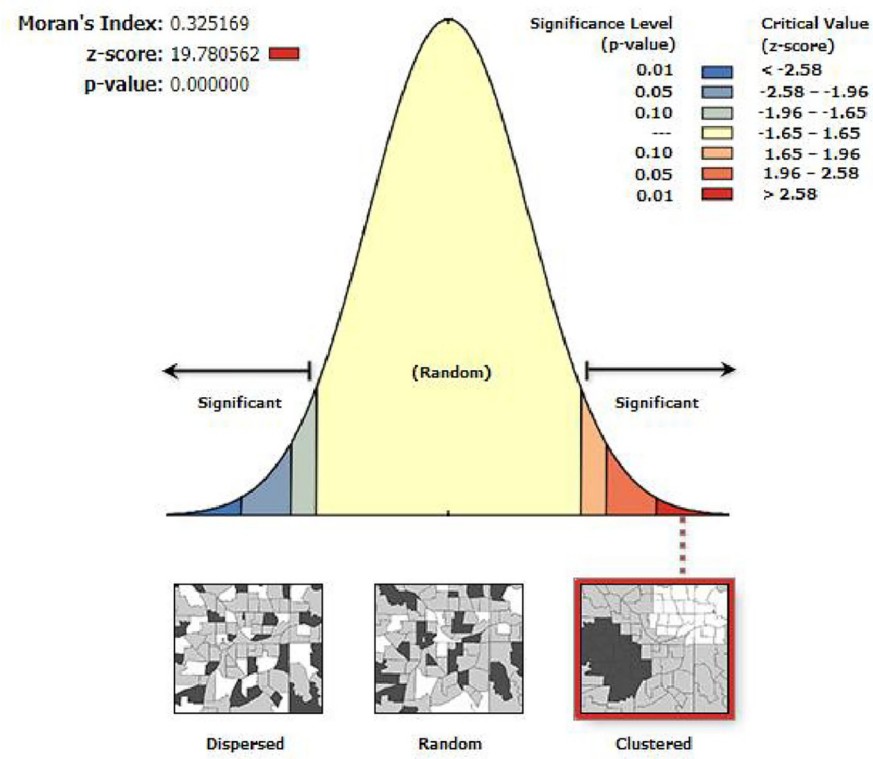

**Fig 3. Spatial autocorrelation analysis of modern contraceptive utilization in Ethiopia, 2016.** Source: Shape file from Ethiopia Central Statistical Agency (CSA).

**Proportion of single women.** Having single marital status decreases modern contraceptive uptake across different regions (Somali, Harari, Dire Dawa, Oromia, SNNPR and Gambela) (Fig 7).

**Proportion of poor women.** Being poor reduced modern contraceptive utilization in different parts of Ethiopia regions (SouthwestAfar, SoutheastAmhara, Southern Beneshangulgumuz, western part of SNNPR, and Central and Northern part of Gambela) (Fig 8).

**Proportion of more fertility preference.** A woman havingmore fertility preference decreases modern contraceptive utilization in Tigray, Southern Oromia, Eastern part of SNNPR, Northern Afar and western Somali regions (Fig 9)

## Multilevel analysis

A multilevel Poisson regression analysis was used to analyze the effect of women's individual characteristics and community-level factors in determining women's use of modern contraceptives. The null model 13.42% (95% CI: 11.39%, 15.76%) of the total variance in the prevalence ratio of modern contraceptive utilization was accounted by between cluster variations of characteristics. The between cluster variability reduced over successive models from 13.42% in the null model into 9.77% (95% CI: 8.10%, 11.75%) in individual-level only model, 9.31% (95% CI: 7.77%, 11.11%) in community-level factors only model and 8.44% (95% CI: 6.98%, 10.17%) in the combined model. Thus, the combined model of individual-level and community-level factors has been preferred for predicting women's modern contraceptive utilization.

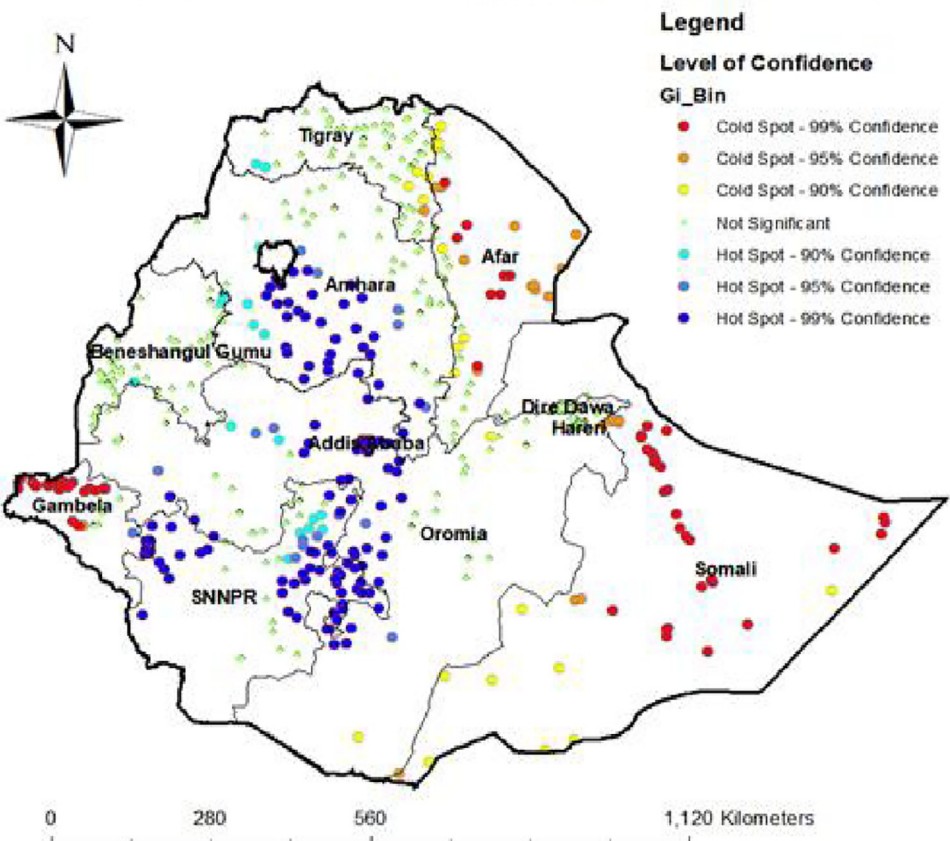

**Fig 4. Hotspot analysis of modern contraceptive utilization in Ethiopia, 2016.** Source: Shape file from Ethiopia Central Statistical Agency (CSA).

### Factors associated with modern contraceptive utilization

First bi-variable Poisson regression analysis was conducted to identify variables that were significant at p-value of 0.25. In bi-variable Poisson regression analysis individual level factors: age, religion, marital status, working status, educational status, wealth index, number of living children, media exposure, fertility preference, and distance to health facility were found significantly associated with modern contraceptive utilization. Community level factors: region, residence, community level wealth index, and community level media exposure were statistically significant factors of modern contraceptive utilization in the bi-variable analysis.

In the multilevel Poisson regression with a robust variance; age, religion, marital status, and fertility preference were found significantly associated with modern contraceptive utilization. Community level factors: region, residence, and community level wealth index were significantly associated with modern contraceptive utilization (Table 5).

The prevalence of modern contraceptive utilization among women aged 25–34 and 35–49 years were decreased by 12% (APR = 0.88, 95% CI: 0.79, 0.98) and 29% (APR = 0.71, 95% CI: 0.61, 0.83) compared to women aged between 15-24years, respectively. Women having married marital status were 2.59 times (APR = 2.59, 95% CI: 2.18, 3.08) higher utilization of modern contraceptive compared to single women. Being others religion followers were decreased

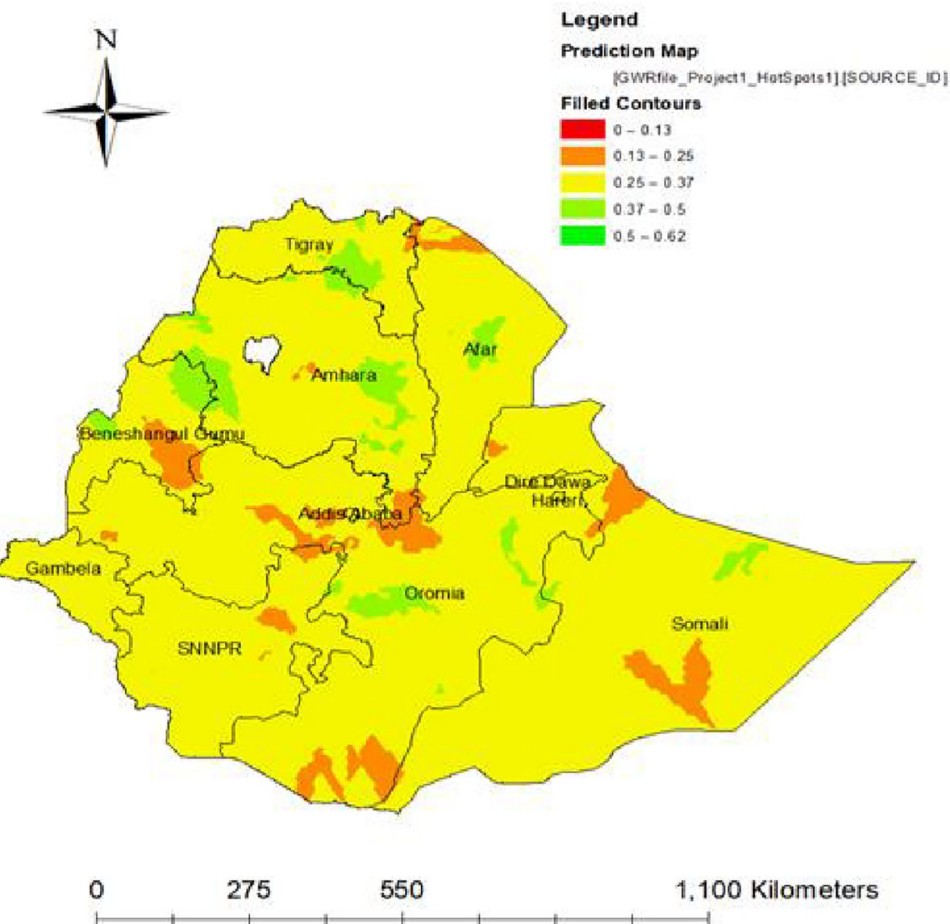

**Fig 5. Spatial interpolation of modern contraceptive utilization in Ethiopia, 2016.** Source: Shape file from Ethiopia Central Statistical Agency (CSA).

utilization of modern contraceptive by 24% (APR = 0.76, 95% CI: 0.65, 0.89) compared to women in Orthodox religion. Number of children 1–4 were increases modern contraceptive use by 1.18 times (APR = 1.18, 95% CI: 1.02, 1.37) compared to women number of children zero. Mothers no more fertility preference had 1.21 times (APR = 1.21, 95% CI: 1.11, 1.32) higher modern contraceptive use than mothers with another fertility preference.

The prevalence of modern contraceptive use was decreased among women living in Afar, Somali, Harari, and Dire Dawa: by 58% (APR = 0.42, 95% CI: 0.27, 0.67), by 94% (APR = 0.06, 95% CI: 0.03, 0.12), by 22% (APR = 0.78, 95% CI: 0.62, 0.98), and by 25% (APR = 0.75, 95% CI: 0.58, 0.98) compared to women living in Tigray region, respectively. Living in Amhara region increases modern contraceptive use by 1.34 times (APR = 1.34, 95% CI: 1.13, 1.57) than living in Tigray region.

Being rural residence decreased modern contraceptive utilization by 20% (APR = 0.80, 95% CI: 0.67, 0.95) compared to urban residence. High community wealth index were decreased modern contraceptive uptake by 22% (APR = 0.78, 95% CI: 0.67, 0.91) compared to low community wealth index (Table 5).

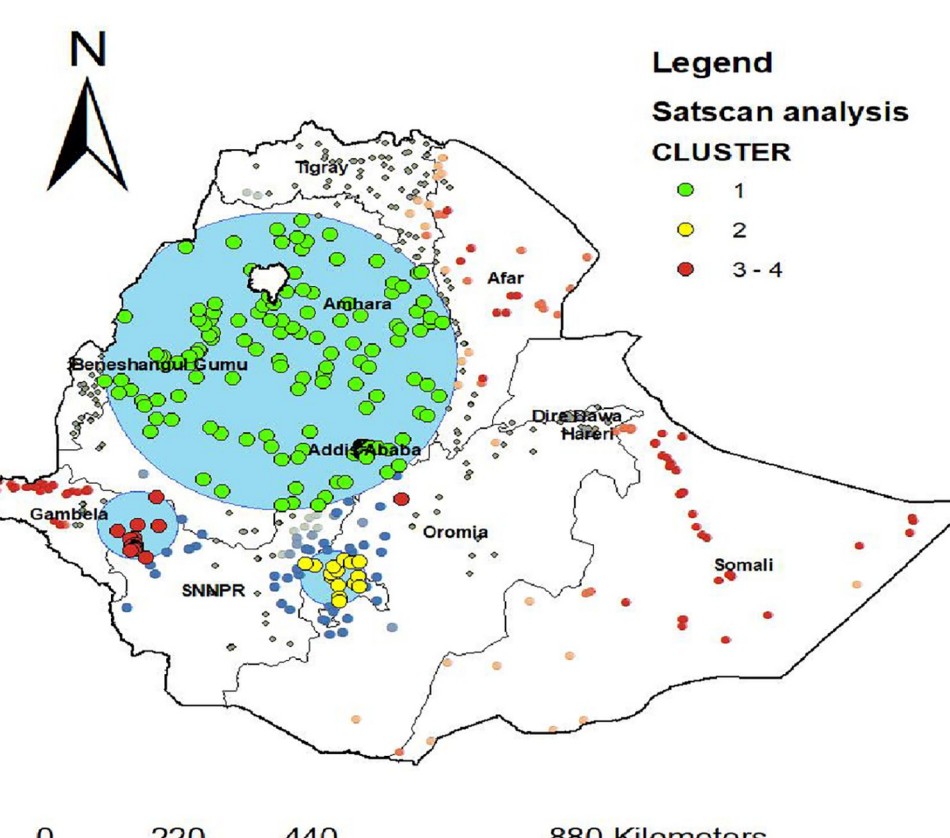

**Fig 6. Spatial scan analysis to detect most likely clusters of modern contraceptive utilization.** Source: Shape file from Ethiopia Central Statistical Agency (CSA).

## Discussion

In this study, we investigated the individual and community-level predictor's association with modern contraceptive use among women in the reproductive age group in Ethiopia and the prevalence of modern contraceptive use was 37.25%. This is higher than a study conducted inMetekel Zone 18.6% [27] and Ghana [1].However, the finding of our study was lower than study conductedinKenya 51% [28]. The possible reasons for this variation could be socio-demographic characteristicsof the study participants. Only 41.9% of respondents in this survey completed a formal education, a lower percentage than Kenya's 93% [28]. Additionally, having access to the media may promote the use of contemporary family planning methods, although in this study, just 23.73% of women have such access [29].

In the spatial regression analysis, marital status, poverty, and fertility preference was significant predictors of hotspot areas of modern contraceptives utilization. An increased proportion of single women decrease the odds of modern contraceptive utilization in Somali, Harari, Dire Dawa, and Oromia regions. The reason might be importance of couple motivation through education, self-reliance of married women, and male participation in reproductive health issues [27, 30].

**Table 3. Ordinary least square (OLS) regression analysis result.**

**Ordinary Least Square (OLS) regression analysis result.**

| Variable | Coefficient | Robust std-error | Robust t-statistics | Robust probability | VIF |
|---|---|---|---|---|---|
| Intercept | 0.677 | 0.053 | 12.859 | 0.000000* | ___ |
| Proportion of women aged 15–24 years | 0.027 | 0.057 | 0.465 | 0.641938 | 1.30 |
| Proportion of women with Muslim religion | 0.054 | 0.029 | 1.831 | 0.067634 | 1.27 |
| Proportion of single women | -0.210 | 0.062 | -3.402 | 0.000728* | 2.03 |
| Proportion of women not currently working | -0.060 | 0.042 | -1.437 | 0.151233 | 1.67 |
| Proportion of women who had no primary education | -0.075 | 0.043 | -1.754 | 0.079983 | 3.33 |
| Proportion of women with poverty | -0.320 | 0.034 | -9.425 | 0.000000* | 2.64 |
| Proportion of women with 0 living children | 0.093 | 0.079 | 1.188 | 0.235369 | 2.34 |
| Proportion of women age $\leq$ 19 years at 1st birth | 0.015 | 0.036 | 0.427 | 0.669379 | 1.02 |
| Proportion of women with low media exposure | 0.037 | 0.046 | 0.808 | 0.419475 | 3.61 |
| Proportion of women want another fertility | -0.269 | 0.044 | -6.142 | 0.000000* | 1.31 |
| Proportion of women with big problem of distance | -0.019 | 0.030 | -0.631 | 0.528291 | 1.77 |

**Ordinary least square regression Diagnostics.**

| | | | | |
|---|---|---|---|---|
| Number of observations | 621 | Adjusted R-squared | | 0.38 |
| Joint F-statistics | 36.17 | Prob(>F), (11,609) degrees of freedom | | 0.000000* |
| Joint Wald statistics | 610.99 | Prob(>chi-squared), (11) degrees of freedom | | 0.000000* |
| Koenker (BP) statistics | 38.39 | Prob(>chi-squared), (11) degrees of freedom | | 0.000067* |
| Jarque–Bera | 8.23 | Prob(>chi-squared), (2) degrees of freedom | | 0.016310* |

**VIF:** Variance Inflation Factor

An increased proportion of more fertility preference decrease the odds of modern contraceptive use in in Tigray, Southern Oromia, Eastern part of SNNPR, Northern Afar and western Somali regions. The possible explanation might be that women who have desire for extra children not use modern contraceptive method to achieve their desires [31]. An increased proportion of poor women reduce the odds of utilizing modern contraceptive in Western Afar, East and Southern Amhara, Southern Beneshangulgumuz, wetern part of SNNPR, and central and Northern Gambela regions. The reason could be, most of the small resources obtained from the petty jobs done by women, and their spouses in poor households are diverted for taking care of the family and less emphasis is placed on the moms' health. Hence, women from poor households refused the service as they encountered difficulties to cover direct and indirect costs incurred in seeking the services [15].

Among the individual level factors, an increase in the age of women was significantly associated with reduce in the use of modern contraceptives. This is similar to results of other studies carried out in Arba-Minch [32], where the utilization of modern contraceptives was negatively influenced by an increase in the age of women. The possible reason might be knowledge gap, beliefs, and attitudes that each woman has. As the age of a woman increases, the probability of changing her attitudes or beliefs towards modern contraception may decrease [33, 34].

**Table 4. Model comparison of OLS and GWR model.**

| Model comparison parameter | OLS model | GWR model |
|---|---|---|
| AICc | -307.87 | -411.61 |
| Adjusted R$^2$ | 0.38 | 0.51 |

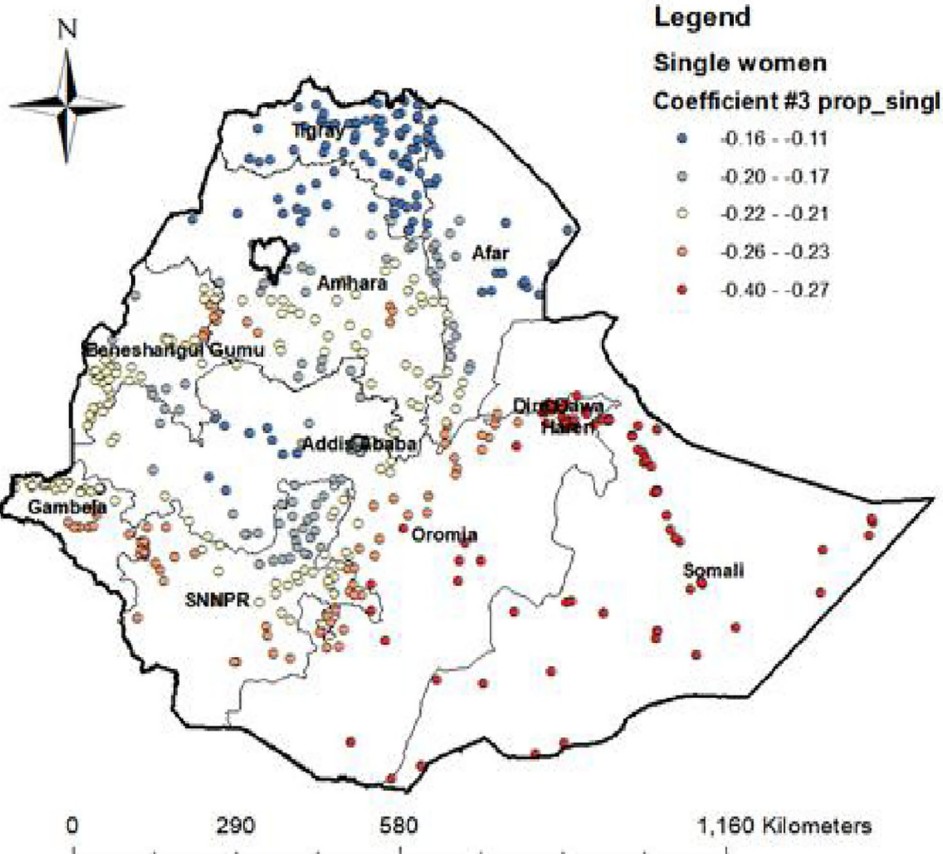

**Fig 7. Single women GWR coefficients for predicting modern contraceptive utilization in Ethiopia, 2016.**

This study revealed that married women had a better tendency to use modern contraception than single women. The result was similar with the studies in Ethiopia [35], Ghana [1], but contradicts with study done in Mali [36]. The possible explanation might be the exposure to maternal and birth control services during antenatal and postnatal cares. Women previous exposure may probably enhance mothers' access to scarce resources and enabled them to use it. Couple motivation is very importance for education, self-reliance of married women, and male participation in reproductive health issues [37].

Others religion followers were found to be less likely to use modern contraceptive methods in Ethiopia. This result is consistent with other study done in Zambia [38]. The possible reason might be due to the fact that religion might have similar socio-cultural importance in influencing the life of women in Ethiopia. Especially, the introduction of some family planning teachings in religiously conservative societies might be disadvantageous [39].

In our study, it was found that, women having living children between 1 and 4, compared with women having no children more likely to use modern contraceptives. This finding is comparable with study report in India [40]. As the number of children increased, women tend to use contraceptive as their desired number of children will be met, while nuliparous women might have had no idea about the use of modern contraceptives [30].

This study found a significant relationship between fertility preference and modern contraceptives utilization. Women who had no desire for more children were more likely to utilize

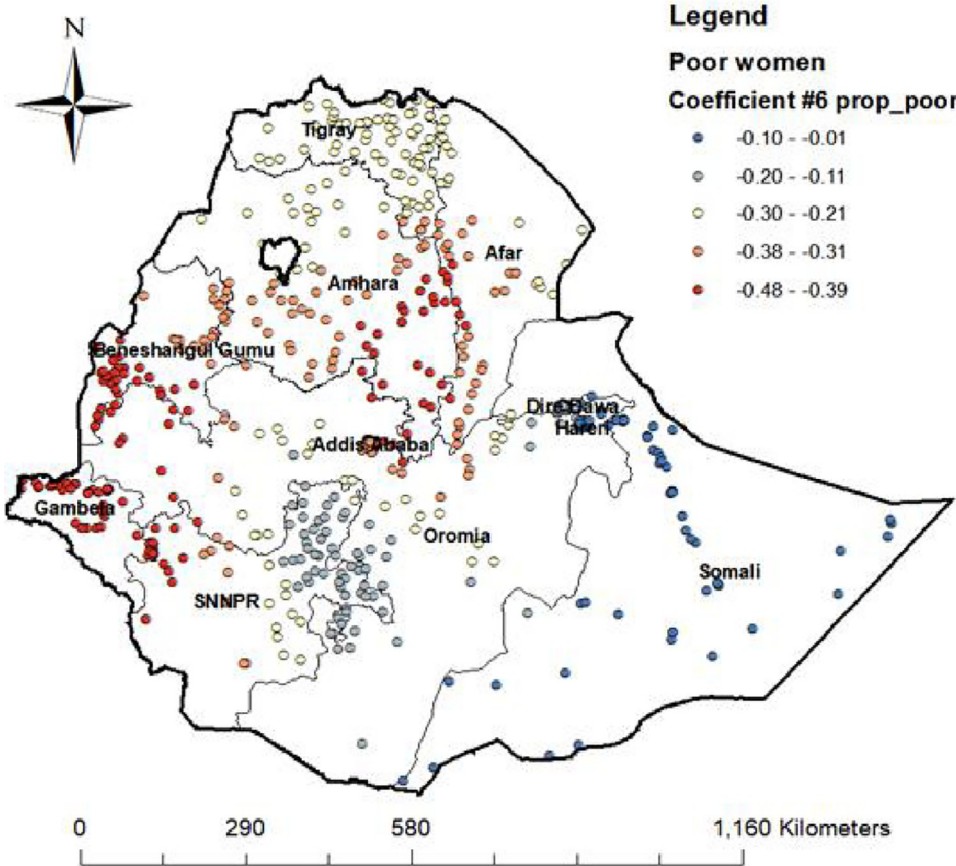

**Fig 8. Poor women GWR coefficients for predicting modern contraceptive utilization in Ethiopia, 2016.**

modern contraceptive than those women who want other children. This finding is in lined with studies conducted in Rwanda [41]. The possible explanation could be that women who have no desire for extra children use modern contraception method to achieve their desires. Since modern contraceptive method is safe, cheap and long term in preventing unwanted pregnancy, it is a method of choice by couples who needs to completely delay childbirth [42].

There is variation of modern contraception utilization across regions and city administrations. Being resident of Amhara region increases modern contraceptive utilization. This variation is confirmed by DHSs report which is conducted every 5years since 2000. Over 16years, between 2000 and 2016, the Amhara region showed an increase in the utilization of modern contraceptives. The large increase in the use of modern contraceptives in the Amhara region might be related to the high number of family planning organizations and the regional government focus on this region [12]. Women from rural settlements are less likely to utilize modern contraception than those from urban settlements. This finding is supported by studies done in Uganda [43]. The possible reason might be low availability and accessibility of the healthcare facilities, trained health care provider, and family planning resources in rural part of Ethiopia. Especially, in pastoral community the problem is more severe in addition to lack of awareness [44].

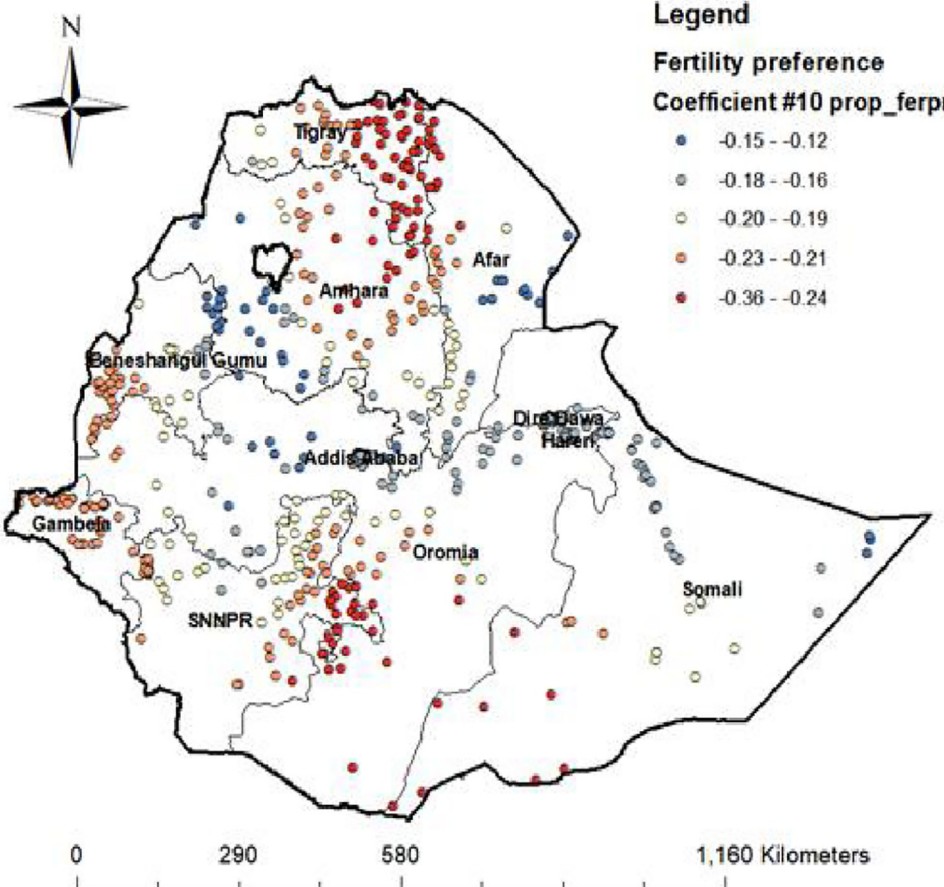

Proportion of fertility preference for predicting modern contraceptive utilization in Ethiopia, 2016

**Fig 9. Want other fertility women GWR coefficients for predicting modern contraceptive utilization in Ethiopia, 2016.**

In this study, high community wealth negatively affects the modern contraceptive use in Ethiopia. Modern contraceptives are available free of charge, the contribution of community wealth for modern contraceptive utilization will not be explained by the ability to pay for the service. Rather it reflects the general socioeconomic position of the community [45]. The reason might be economically matured communities are getting modern contraceptive services from dispensary rather than regular MCH department, this results under report of modern contraception among high economic class [19].

## Strength and limitation

Large sample sizes and the use of nationally representative data may have improved our ability to estimate the parameters. To account for the hierarchical structure of the data from the EDHS, the study also used a multilevel analysis. Similar to this, it was crucial to perform spatial analysis and GWR to determine the geographic variation and, separately, the predictors of modern contraceptive use. Policy makers will benefit from this study's assistance in developing or improving intervention methods based on the identified spatial variances.

**Table 5. Multilevel analysis of modern contraceptive utilization in Ethiopia, EDHS 2016.**

| Variables | Null model | Model I | Model II | Model III |
|---|---|---|---|---|
| **Characteristics** | | Individual level | Community level | Individual and |
| | | characteristics | characteristics | community |
| | | | | characteristics |
| | | APR (95% CI) | APR (95% CI) | APR (95% CI) |
| **Age** | | | | |
| **15–24** | | 1 | | 1 |
| **35–34** | | 0.91(0.82, 1.01) | | 0.88(0.79, 0.98)* |
| **35–49** | | 0.75(0.64, 0.87)** | | 0.71(0.61, 0.83)** |
| **Educational status** | | | | |
| **no education** | | 1 | | 1 |
| **primary** | | 1.02(0.93, 1.12) | | 1.00(0.91, 1.10) |
| **secondary and above** | | 1.05(0.93, 1.19) | | 1.01(0.89, 1.14) |
| **Marital status** | | | | |
| **single** | | 1 | | 1 |
| **married** | | 2.55(2.15, 3.03)** | | 2.59(2.18, 3.08)** |
| **Religion** | | | | |
| **orthodox** | | 1 | | 1 |
| **protestant** | | 0.98(0.74, 1.29) | | 1.07(0.81, 1.42) |
| **muslim** | | 1.06(0.94, 1.19) | | 1.14(0.99, 1.32) |
| **others** | | 0.62(0.53, 0.62)** | | 0.76(0.65, 0.89)** |
| **No. living children** | | | | |
| **0** | | 1 | | 1 |
| **1_4** | | 1.17(1.01, 1.35)* | | 1.18(1.02, 1.37)* |
| **≥5** | | 1.03(0.84, 1.26) | | 1.09(0.89, 1.34) |
| **Fertility preference** | | | | |
| **want another** | | 1 | | 1 |
| **undecided** | | 0.96(0.76, 1.21) | | 0.94(0.75, 1.18) |
| **want no more** | | 1.25(1.15, 1.36)** | | 1.21(1.11, 1.32)** |
| **working status** | | | | |
| **no** | | 1 | | 1 |
| **yes** | | 1.10(1.00, 1.20)* | | 1.09(0.99, 1.20) |
| **Wealth index** | | | | |
| **poor** | | 1 | | 1 |
| **middle** | | 1.20(1.05. 1.37)** | | 1.13(0.99, 1.30) |
| **rich** | | 1.23(1.09, 1.39)** | | 1.11(0.98, 1.26) |
| **Distance to health facility** | | | | |
| **big problem** | | 1 | | 1 |
| **not problem** | | 1.06(0.97, 1.15) | | 1.01(0.93, 1.10) |
| **Media exposure** | | | | |
| **low** | | 1 | | 1 |
| **high** | | 1.05(0.95, 1.16) | | 1.02(0.92, 1.14) |
| **Region** | | | | |
| **Tigray** | | | 1 | 1 |
| **Afar** | | | 0.36(0.22, 0.57)** | 0.42(0.27, 0.67)** |
| **Amhara** | | | 1.32(1.12, 1.56)** | 1.34(1.13, 1.57)** |
| **Oromia** | | | 0.83(0.67, 1.03) | 0.82(0.66, 1.01) |
| **Somali** | | | 0.04(0.02, 0.09)** | 0.06(0.03, 0.12)** |

(*Continued*)

**Table 5.** (Continued)

| Variables | Null model | Model I | Model II | Model III |
|---|---|---|---|---|
| **Benishangul** | | | 0.92(0.74, 1.15) | 0.88(0.70, 1.11) |
| **SNNPR** | | | 1.27(1.05, 1.54)* | 1.06(0.86, 1.29) |
| **Gambela** | | | 0.84(0.66, 1.08) | 0.76(0.58, 1.01) |
| **Harari** | | | 0.72(0.58, 0.89)** | 0.78(0.62, 0.98)* |
| **Addis Ababa** | | | 0.81(0.67, 0.98)* | 0.91(0.78, 1.08) |
| **Dire dawa** | | | 0.65(0.50, 0.84)** | 0.75(0.58, 0.98)* |
| **Residence** | | | | |
| **urban** | | | 1 | 1 |
| **rural** | | | 0.80(0.66, 0.97)* | 0.80(0.67, 0.95)* |
| **Community wealth index** | | | | |
| **low** | | | 1 | 1 |
| **high** | | | 0.72(0.61, 0.84)** | 0.78(0.67, 0.91)** |
| **Community media exposure** | | | | |
| **low** | | | 1 | 1 |
| **high** | | | 1.10(0.93, 1.31)** | 1.10(0.94, 1.28) |
| **Model comparison and** | | | | |
| **random effect** | | | | |
| **ICC** | 0.134(0.11, 0.15) | _ | _ | _ |
| **Deviance** | 19,978.47 | 19,261.88 | 19,667.04 | 19,039.41 |
| **AIC** | 23,153.31 | 22,518.60 | 22,984.78 | 22,435.00 |

*P-value < 0.05

**P-value <0.01

Big problem: Distance from home to health facility took >30min on foot

Not big problem: Distance from home to health facility took <30min on foot

Cross-sectional data were employed in this analysis, which limits the inferences about the causes of the factors on the dependent variable. Additionally, several variables were not taken into consideration since they were not present in the EDHS data set.

## Conclusion and recommendation

There is a significant spatial variation of factors affecting modern contraceptive use across regions in Ethiopia. Therefore, public health interventions targeting areas with low modern contraceptive utilization will help to increase modern contraception use considering significant factors at individual and community levels. Locations with low modern contraceptive use and its predictors could assist program planners and decision-makers to design targeted public health interventions.

Government of Ethiopia must develop more geographic targeted strategies for improving socioeconomic status of women and availability & accessibility of health facilities in rural areas of the countries. This will not only increase modern contraceptive provision, but will also reduce teenage pregnancy and birth, and in turn, contribute to the achievement of the Sustainable Development Goal three. Improving modern contraception use among reproductive age group will also require connecting women with information and services during their routine health service visits and taking advantage of missed opportunities for contact with the health facility.

## Supporting information

**S1 File.**
(DTA)

## Acknowledgments

We greatly acknowledge MEASURE DHS for granting access to the EDHS data sets.

## Author Contributions

**Conceptualization:** Yazachew Moges Chekol, Getayeneh Antehunegn Tesema.

**Data curation:** Yazachew Moges Chekol.

**Formal analysis:** Yazachew Moges Chekol, Bazezew Takel Goshe, Lewi Goytom Gebrehewet.

**Methodology:** Yazachew Moges Chekol, Setotaw Begashaw Jemberie, Bazezew Takel Goshe, Getayeneh Antehunegn Tesema, Zemenu Tadesse Tessema, Lewi Goytom Gebrehewet.

**Software:** Yazachew Moges Chekol, Setotaw Begashaw Jemberie, Bazezew Takel Goshe, Lewi Goytom Gebrehewet.

**Supervision:** Getayeneh Antehunegn Tesema, Zemenu Tadesse Tessema.

**Validation:** Getayeneh Antehunegn Tesema, Zemenu Tadesse Tessema.

**Visualization:** Yazachew Moges Chekol, Setotaw Begashaw Jemberie, Lewi Goytom Gebrehewet.

**Writing – original draft:** Yazachew Moges Chekol.

**Writing – review & editing:** Getayeneh Antehunegn Tesema, Zemenu Tadesse Tessema.

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
