## [Decision Letter · Decision Letter 0]

26 Oct 2022

PONE-D-22-10071Geographic weighted regression analysis of hotspots of modern contraceptive utilization and its associated factors in Ethiopia

PLOS ONE

Dear Dr. Chekol,

Thank you for submitting your manuscript to PLOS ONE. After careful consideration, we feel that it has merit but does not fully meet PLOS ONE’s publication criteria as it currently stands. Therefore, we invite you to submit a revised version of the manuscript that addresses the points raised during the review process.

Your manuscript has been assessed by one peer-reviewer and their report is appended below. 

 The reviewer comments that your methodology section requires further clarification and that your tables should report confidence intervals in the percentage column. In addition, the reviewer comments that the discussion second of your study requires more information, and the conclusions need to be reported better. 

 In addition to the reviewer's comments, it has come to the editorial office's attention that there appear to be a lot of grammar, spelling, and typological errors in your manuscript. For example, the abstract reads "Therefore, the this study aimed to investigate" as well as "public health interventions targeting colds pot areas of modern contraceptive considering rural residents". Please make sure to carefully review the writing of the manuscript. 

 Furthermore, it has come to the editorial office's attention that in Editorial Manager, the submission details of your manuscript only list one author, whereas your title page suggests 6 authors contributed to this article. Please make sure that you register each author on your manuscript correctly. Please note that the journal will follow up with each author individually to confirm that they know of, and consent to the submission of this manuscript. 

Please note that we have only been able to secure a single reviewer to assess your manuscript. We are issuing a decision on your manuscript at this point to prevent further delays in the evaluation of your manuscript. Please be aware that the editor who handles your revised manuscript might find it necessary to invite additional reviewers to assess this work once the revised manuscript is submitted. However, we will aim to proceed on the basis of this single review if possible. 

We look forward to receiving your revised manuscript.

Kind regards,

Maria Elisabeth Johanna Zalm, Ph.D

Editorial Office

PLOS ONE

Journal Requirements:

4. Please amend the manuscript submission data (via Edit Submission) to include author Setotaw Begashaw Jemberie, Bazezew Takel Goshe, Getayeneh Antehunegn Tesema, Zemenu Tadesse Tessema and Lewi Goytom Gebrehewet.

5. We note that Figure 4, 6, 7, 8, 9, 10 and 11 in your submission contain [map/satellite] images which may be copyrighted. All PLOS content is published under the Creative Commons Attribution License (CC BY 4.0), which means that the manuscript, images, and Supporting Information files will be freely available online, and any third party is permitted to access, download, copy, distribute, and use these materials in any way, even commercially, with proper attribution. For these reasons, we cannot publish previously copyrighted maps or satellite images created using proprietary data, such as Google software (Google Maps, Street View, and Earth). For more information, see our copyright guidelines: http://journals.plos.org/plosone/s/licenses-and-copyright.

a. You may seek permission from the original copyright holder of Figure 4, 6, 7, 8, 9, 10 and 11 to publish the content specifically under the CC BY 4.0 license.  

6. Please upload a copy of Figure S1, to which you refer in your text. If the figure is no longer to be included as part of the submission please remove all reference to it within the text.

7. Thank you for submitting the above manuscript to PLOS ONE. During our internal evaluation of the manuscript, we found significant text overlap between your submission and the following previously published works, some of which you are an author.

- https://bmjopen.bmj.com/content/10/10/e037532.full

- https://bmjopen.bmj.com/content/10/9/e041103.full

Please revise the manuscript to rephrase the duplicated text, cite your sources, and provide details as to how the current manuscript advances on previous work. Please note that further consideration is dependent on the submission of a manuscript that addresses these concerns about the overlap in text with published work.

Reviewers' comments:

Reviewer's Responses to Questions

**Comments to the Author**

1. Is the manuscript technically sound, and do the data support the conclusions?

Reviewer #1: Partly

2. Has the statistical analysis been performed appropriately and rigorously? 

Reviewer #1: Yes

3. Have the authors made all data underlying the findings in their manuscript fully available?

Reviewer #1: Yes

4. Is the manuscript presented in an intelligible fashion and written in standard English?

Reviewer #1: No

5. Review Comments to the Author

Reviewer #1: Reviewer report

Introduction

Line 118-123… it could be good if this paragraph shows what are the gaps in the literature and what would this research is intended to address.

Methods

Line 151.. ‘…The study population was all Women who had any modern contraceptive in the five years…’ it is not clear; are you considered only those who had used contraceptive in the last five years?

The inclusion and exclusion criteria were not clear. Did you include or exclude sexually active women?

Line 197… what is the national median value?

Line 200… what are your bases to define Media exposure in such a way? it is the standard definition? If so, cite a reference

Line 240… one of the issues with GWR is multicollinearity, did the authors check multicollinearity?

Results

Table 1.. which religions are grouped under the ‘other’ category? Add confidence intervals for the percentage column.

What is the relevance of figure 3 as the values already available in table 2?

Figure 6 reported three confidence levels (90%, 95%, and 99%), I suggest using only one confidence level.

Line 400-415. I would suggest restating this paragraph as the authors used the one phrase ‘… there is negative relationship between proportion of…’ again and again.

Table 5… how did you define /measure the distance from the health facility? What does big problem and not problem mean?

Table 5… In EDHS there are five categories for wealth index. Why did you have 3 categories of wealth index?

Table 5. Why Tigray region selected as a reference?

Discussion

Line 525… “The highest modern contraceptive utilization was recorded in the Amhara region…”. This statement contradicts your result- contraceptive prevalence was higher in Oromia region (37%) than Amhara region (25%)

Line 507-511… “…. Other religion followers were found to be less likely to use modern contraceptive methods in Ethiopia. This result is consistent with other study done in Zambia[43]. The possible reason might be due to the fact that religion might have similar socio-cultural importance in influencing the life of women in Ethiopia. Especially, the introduction of some family planning teachings in religiously conservative societies might be disadvantageous……”.

This paragraph is not clear. Which religions were grouped under ‘other region followers’? Be cautious to compare these variables in Ethiopia with Zambia.

Conclusion

What is your basis to conclude the prevalence of modern contraceptives in Ethiopia was low?

It is not informative stating as a hotspot and cold spot, as these are not defined before.

6. PLOS authors have the option to publish the peer review history of their article (what does this mean?). If published, this will include your full peer review and any attached files.

Reviewer #1: No

---

## [Author Response · Author response to Decision Letter 0]

21 Jan 2023

Response to Reviewers

Reviewers' comments: 

The reviewer comments that your methodology section requires further clarification and that your tables should report confidence intervals in the percentage column. In addition, the reviewer comments that the discussion second of your study requires more information, and the conclusions need to be reported better. 

 Author’s Responses to comments

Extensive revisions were made in methodology, discussion and conclusion sections. 

Table 1&2 indicates socio-demographic characteristics or descriptive analysis of the study participants. So, no need of writing confidence interval in the percentage column. But confidence intervals were reported in the statistical analysis or in the multilevel analysis table.

Reviewers' comments: 

In addition to the reviewer's comments, it has come to the editorial office's attention that there appear to be a lot of grammar, spelling, and typological errors in your manuscript. For example, the abstract reads "Therefore, the this study aimed to investigate" as well as "public health interventions targeting colds pot areas of modern contraceptive considering rural residents". Please make sure to carefully review the writing of the manuscript. 

 Author’s Responses to comments

Grammar, spelling, and typological error were observed significantly in our manuscript and we tried to correct as far as our knowledge. 

Reviewers' comments: 

Furthermore, it has come to the editorial office's attention that in Editorial Manager, the submission details of your manuscript only list one author, whereas your title page suggests 6 authors contributed to this article. Please make sure that you register each author on your manuscript correctly. Please note that the journal will follow up with each author individually to confirm that they know of, and consent to the submission of this manuscript. 

Author’s Responses to comments 

Only corresponding authors were registered mistakenly in our previous submission. But all authors were registered in our revised manuscript submission. 

Reviewers' comments: 

5. We note that Figure 4, 6, 7, 8, 9, 10 and 11 in your submission contain [map/satellite] images which may be copyrighted. All PLOS content is published under the Creative Commons Attribution License (CC BY 4.0), which means that the manuscript, images, and Supporting Information files will be freely available online, and any third party is permitted to access, download, copy, distribute, and use these materials in any way, even commercially, with proper attribution. For these reasons, we cannot publish previously copyrighted maps or satellite images created using proprietary data, such as Google software (Google Maps, Street View, and Earth). For more information, see our copyright guidelines: http://journals.plos.org/plosone/s/licenses-and-copyright.

Author’s Responses to comments

Dear reviewers, we would like to assure you that all figures in our manuscript is our original work which are produced using our data set in ArcGIS version 10.7 and SaTScan version 9.6 software and it is a part of spatial epidemiology course which is our specialization area as an epidemiologist. So, figures are not copyrighted or satellite image.

As we stated in the methodology part “The survey data was accessed from the measure Demographic and Health Survey (https://dhsprogram.com/).” Only EDHS 2016 data set with it is shape file was requested and received online using (https://dhsprogram.com/) this address.

Reviewers' comments: 

6. Please upload a copy of Figure S1, to which you refer in your text. If the figure is no longer to be included as part of the submission please remove all reference to it within the text.

Author’s Responses to comments

All reference were removed from our text because the figure is no longer to be included as part of our manuscript submission.

Reviewers' comments: 

7. Thank you for submitting the above manuscript to PLOS ONE. During our internal evaluation of the manuscript, we found significant text overlap between your submission and the following previously published works, some of which you are an author.

Author’s Responses to comments

Significant text overlapping was observed during internal evaluation of our manuscript and revisions were made in our manuscript by rephrasing the duplicated text. Furthermore, ideas were incorporated that explain how the current manuscript advances on previous studies.

 Reviewers' comments: 

Reviewer #1: Reviewer report

Introduction

Line 118-123… it could be good if this paragraph shows what are the gaps in the literature and what would this research is intended to address.

Author’s Responses to comments

This paragraph (line 118-123) already replaced by paragraph that shows the gaps in the previous literature and the gaps intended to be addressed by this study. 

 Reviewers' comments: 

Methods

Line 151.. ‘…The study population was all Women who had any modern contraceptive in the five years…’ it is not clear; are you considered only those who had used contraceptive in the last five years?

The inclusion and exclusion criteria were not clear. Did you include or exclude sexually active women?

Author’s Responses to comments

Study population, inclusion and exclusion criteria were cleared with revision based on our study participant selection.

 Reviewers' questions: 

Line 197… what is the national median value?

Author’s Responses to comments

Ethiopian Demographic Health Survey (EDHS) were considered as representative data at national level. Since, we don’t have any standard to classify the community wealth status as high and low community poverty; we generated the national median value (.3333333) using the household wealth index in each enumeration areas. After checking the normality of the distribution the median value was taken to classify high/low poverty status of the community, since the distribution is not normal. 

 Reviewers' questions: 

Line 200… what are your bases to define Media exposure in such a way? it is the standard definition? If so, cite a reference

Author’s Responses to comments

Operational definition of media exposure was taken from Ethiopian Demographic Health Survey (EDHS) manual, which is considered as standard definition and the reference cited.

 Reviewers' questions: 

Line 240… one of the issues with GWR is multicollinearity, did the authors check multicollinearity?

Author’s Responses to comments

Multicollinearity was checked and already reported under table 3 during Geographic Weighted Regression (GWR) analysis using variance Inflation Factor (VIF) as a default result reported during GWR analysis in ArcGIS software. 

 Reviewers' questions: 

Results

Table 1, which religions are grouped under the ‘other’ category? Add confidence intervals for the percentage column.

Author’s Responses to comments

Religions under other religion categories are: Catholic and traditional

Table 1&2 indicates socio-demographic characteristics or descriptive analysis of the study participants. So, no need of writing confidence interval in the percentage column. But confidence intervals were reported in the statistical analysis or in the multilevel analysis table.

 Reviewers' questions: 

What is the relevance of figure 3 as the values already available in table 2?

Author’s Responses to comments

Comment accepted and figure 3 was removed since it is redundant 

 Reviewers' questions: 

Figure 6 reported three confidence levels (90%, 95%, and 99%), I suggest using only one confidence level.

Author’s Responses to comments

Three confidence levels are a default in ArcGIS software to report hot spot analysis. But the readers can use their preference level of confidence interval. 

 Reviewers' questions: 

Line 400-415. I would suggest restating this paragraph as the authors used the one phrase ‘… there is negative relationship between proportion of…’ again and again.

Author’s Responses to comments

Comment accepted and the paragraph restated 

 Reviewers' questions: 

Table 5… how did you define /measure the distance from the health facility? What does big problem and not problem mean?

Author’s Responses to comments

Distance from home to health facility took >30min on foot considered as big problem and distance from home to health facility took <30min on foot considered as not big problem

 Reviewers' questions: 

Table 5… In EDHS there are five categories for wealth index. Why did you have 3 categories of wealth index?

Author’s Responses to comments

Most of the time many scholars use three categories of wealth index, in Ethiopian context five categories of wealth index doesn’t give sense. For example, the gap between poorest and poorer, richer and richest is not well understandable in our community. So, authors prefer to use three categories of wealth index not to confuse our readers. 

 Reviewers' questions: 

Table 5. Why Tigray region selected as a reference?

Author’s Responses to comments

Tigray region selected as reference because of better health infrastructure and better health service coverage in the country. 

 Reviewers' questions: 

Discussion

Line 525… “The highest modern contraceptive utilization was recorded in the Amhara region…”. This statement contradicts your result- contraceptive prevalence was higher in Oromia region (37%) than Amhara region (25%)

Author’s Responses to comments

This was typing error and already corrected 

 Reviewers' questions: 

Line 507-511… “…. Other religion followers were found to be less likely to use modern contraceptive methods in Ethiopia. This result is consistent with other study done in Zambia[43]. The possible reason might be due to the fact that religion might have similar socio-cultural importance in influencing the life of women in Ethiopia. Especially, the introduction of some family planning teachings in religiously conservative societies might be disadvantageous……”.

This paragraph is not clear. Which religions were grouped under ‘other region followers’? Be cautious to compare these variables in Ethiopia with Zambia.

Author’s Responses to comments

List of religions under “other” categories are Catholic and traditional as it’s explained previously. When we compare these variables between Ethiopia and Zambia, we understand that from different literatures there is similarity between two countries community in the aspect of religion. 

 Reviewers' questions: 

Conclusion

What is your basis to conclude the prevalence of modern contraceptives in Ethiopia was low?

It is not informative stating as a hotspot and cold spot, as these are not defined before.

Author’s Responses to comments

The current prevalence of modern contraception in Ethiopia was low as we compared with the national target which is should be >80%.

---

## [Decision Letter · Decision Letter 1]

6 Jun 2023

PONE-D-22-10071R1Geographic weighted regression analysis of hot spots of modern contraceptive utilization and its associated factors in Ethiopia: A geographic weighted regression analysis and multilevel robust Poisson regression analysisPLOS ONE

Dear Dr. Chekol,

Thank you for submitting your manuscript to PLOS ONE. After careful consideration, we feel that it has merit but does not fully meet PLOS ONE’s publication criteria as it currently stands. Therefore, we invite you to submit a revised version of the manuscript that addresses the points raised during the review process.

We look forward to receiving your revised manuscript.

Kind regards,

Samuel Hailegebreal

Academic Editor

PLOS ONE

Journal Requirements:

Additional Editor Comments:

reviewer #3

Thank you for allowing me to review this paper. The paper is addressed important public health topic. I have provided some question and comments as outlined below.

General question

1. What is the need of doing another research since there is a similar study done in Ethiopia?

2. What will it add to the already known facts?

3. What was the gap in the previous study?

4. Why you select multilevel robust Poisson analysis for this study?

General comment

I. Title: In the title you should remove the methods that you used

II. Introduction section: I would like to thank you, your introduction is very attractive and well written. While, it is better to include consequences of lack of modern contraceptive utilization.

III. Method section: it is too lengthy & try to summarized sentence especially, your method of analysis like spatial analysis part. What are the assumption that you have checked In OLS model? Please write shortly and state in method section.

IV. Result section: Dhs measure education as o education, primary, secondary, and higher. Why did you merge secondary ad higher? You could have merged it only if the distribution of one was extremely small, which is not the case. In addition, did you focus only on married women? If you did not, what about women who were single?

From line number 306-312 already reported in the hotspot analysis section, so please try to remove this paragraph

A beauty of this study is the large data and the multilevel analysis performed. However, you did not include in the background, the importance of the community factors that warranted a multilevel analysis. Overall, the discussion is shallow and is a repetition of the results. Furthermore, it needs some work on language.

Reviewers' comments:

Reviewer's Responses to Questions

**Comments to the Author**

1. If the authors have adequately addressed your comments raised in a previous round of review and you feel that this manuscript is now acceptable for publication, you may indicate that here to bypass the “Comments to the Author” section, enter your conflict of interest statement in the “Confidential to Editor” section, and submit your "Accept" recommendation.

Reviewer #2: All comments have been addressed

Reviewer #3: (No Response)

2. Is the manuscript technically sound, and do the data support the conclusions?

Reviewer #2: Yes

Reviewer #3: Yes

3. Has the statistical analysis been performed appropriately and rigorously? 

Reviewer #2: Yes

Reviewer #3: Yes

4. Have the authors made all data underlying the findings in their manuscript fully available?

Reviewer #2: Yes

Reviewer #3: Yes

5. Is the manuscript presented in an intelligible fashion and written in standard English?

Reviewer #2: Yes

Reviewer #3: Yes

6. Review Comments to the Author

Reviewer #2: Proof read the whole document for better readability. Regarding, the model some variables like community level wealth index is total methodologically wrong and there is no community level wealth index. If possible check it again, what model has been used to aggregate individual level characteristics to area level.

Reviewer #3: (No Response)

7. PLOS authors have the option to publish the peer review history of their article (what does this mean?). If published, this will include your full peer review and any attached files.

Reviewer #2: No

Reviewer #3: No

---

## [Author Response · Author response to Decision Letter 1]

11 Jun 2023

Response to Reviewers

Reviewers' comments: 

 Journal Requirements: 

Author’s Response 

Dear Sir/Madam, I would like to assure you that my reference is complete and correct also no cited paper that has been retracted. 

Reviewer question

1. What is the need of doing another research since there is a similar study done in Ethiopia?

Author’s Response

The previous study was conducted only considering married women as a study population and without considering of geographical variation of variables across Ethiopian regions. But there is evidence that indicates variation of variables influence among regions in health service utilization and unmarried women are sexually active almost the same as married women[20]. Therefore, this study aimed to investigate spatial regression analysis of modern contraceptives use among women of reproductive age in Ethiopia, irrespective of their marital status and identify the potential factors associated with the use of modern contraceptives and considering geographic variation of variables. As a result this study result will facilitate evidence based decision making by complementing limitations of the previous studies. 

Reviewer question

2. What will it add to the already known facts?

Author’s Response

 Findings of this study will be useful for health planners, policymakers, and non-governmental partners who are working to improve the health and well-being of women of reproductive age in Ethiopia. Besides, mapping hotspot areas of modern contraceptive use, it will provide a deeper understanding of the impacts of already implemented interventions in each region of the country. Furthermore, it will assist in designing programs and strategies to increase coverage, quality, and equity of modern contraceptive service utilization and it is determinant factors at regional level.

Reviewer question

3. What was the gap in the previous study?

Author’s Response

The previous study was conducted only considering married women as a study population and without considering of variation of variables across Ethiopian regions (Geographical weighted regression analysis not done).

Reviewer question

4. Why you select multilevel robust Poisson analysis for this study?

Author’s Response

In our study the ICC shows that there is a significant clustering effect (ICC>10%). In this case, the prevalence ratio is the best measures of association because reporting the odds ratio will result over estimation of association between modern contraception use and it is predictors.

Reviewer general comment 

I. Title: In the title you should remove the methods that you used

Author’s Response

Dear reviewer, I thank you for your constructive comment and already I have removed it 

II. Introduction section: I would like to thank you; your introduction is very attractive and well written. While, it is better to include consequences of lack of modern contraceptive utilization.

Author’s Response

Dear reviewer, first of all, thank you for your detail review and comment, but consequences of lack of modern contraceptive utilization was mentioned between line number 87 and 91 as a positive statement or “the advantage of the modern contraceptive use among reproductive age is very vital for designing interventions, plans, and policies to address premature age pregnancies and other related issues. It is also useful to reduce unsafe abortions, maternal death, and sexually transmitted infections (STI)”. I left listing consequences of lack of modern contraceptive utilization intentionally in order to avoid idea redundancy. 

Reviewer general comment 

III. Method section: it is too lengthy & try to summarized sentence especially, your method of analysis like spatial analysis part. What are the assumptions that you have checked In OLS model? Please write shortly and state in method section.

Author’s Response

Accepted and corrected

Reviewer general comment 

IV. Result section: Dhs measure education as o education, primary, secondary, and higher. Why did you merge secondary ad higher? You could have merged it only if the distribution of one was extremely small, which is not the case. In addition, did you focus only on married women? If you did not, what about women who were single? From line number 306-312 already reported in the hotspot analysis section, so please try to remove this paragraph A beauty of this study is the large data and the multilevel analysis performed. However, you did not include in the background, the importance of the community factors that warranted a multilevel analysis. Overall, the discussion is shallow and is a repetition of the results. Furthermore, it needs some work on language.

Author’s Response

 Number of participants with higher educational status is extremely low and failed to satisfy the Chi-square assumption that why we decide to merge them with secondary education. As we have clearly stated under the inclusion criteria all Women in the reproductive age group were included in the study irrespective of their marital status and this is one of the uniqueness of our study from previous studies. 

Dear reviewer, result reported from line number 306-312 and ideas reported under in the hotspot analysis section are different. OLS regression results were reported from line number 306-312 and may be regression results consistent with hotspot analysis since it is statistical evidence. So, it is statistical evidence that strengthening our hot spot analysis and indicating our result is consistent. 

Sentence which states about the importance of the community level factors that warranted a multilevel analysis included under introduction section

---

## [Decision Letter · Decision Letter 2]

4 Jul 2023

Geographic weighted regression analysis of hot spots of modern contraceptive utilization and its associated factors in Ethiopia

PONE-D-22-10071R2

Dear Dr. Chekol,

We’re pleased to inform you that your manuscript has been judged scientifically suitable for publication and will be formally accepted for publication once it meets all outstanding technical requirements.

Kind regards,

Samuel Hailegebreal

Academic Editor

PLOS ONE

Additional Editor Comments (optional):

Reviewers' comments:

Reviewer's Responses to Questions

**Comments to the Author**

1. If the authors have adequately addressed your comments raised in a previous round of review and you feel that this manuscript is now acceptable for publication, you may indicate that here to bypass the “Comments to the Author” section, enter your conflict of interest statement in the “Confidential to Editor” section, and submit your "Accept" recommendation.

Reviewer #3: All comments have been addressed

2. Is the manuscript technically sound, and do the data support the conclusions?

Reviewer #3: Yes

3. Has the statistical analysis been performed appropriately and rigorously? 

Reviewer #3: Yes

4. Have the authors made all data underlying the findings in their manuscript fully available?

Reviewer #3: Yes

5. Is the manuscript presented in an intelligible fashion and written in standard English?

Reviewer #3: Yes

6. Review Comments to the Author

Reviewer #3: Thank you, All my comments have been addressed.For the future work your method of analysis like spatial analysis part. Please write shortly and separated manner in method section

7. PLOS authors have the option to publish the peer review history of their article (what does this mean?). If published, this will include your full peer review and any attached files.

Reviewer #3: No

---

## [Editor Report · Acceptance letter]

17 Jul 2023

PONE-D-22-10071R2 

Geographic weighted regression analysis of hot spots of modern contraceptive utilization and its associated factors in Ethiopia 

Dear Dr. Chekol:

I'm pleased to inform you that your manuscript has been deemed suitable for publication in PLOS ONE. Congratulations! Your manuscript is now with our production department. 

Kind regards, 

on behalf of

Mr. Samuel Hailegebreal 

Academic Editor

PLOS ONE